

# Frequency-independent optical spin injection in Weyl semimetals

Yang Gao[1,2], Chong Wang[3,4] and Di Xiao[3,4*]

**1** Department of Physics, University of Science and Technology of China,
Hefei, Anhui 230026, China
**2** ICQD, Hefei National Laboratory for Physical Sciences at Microscale,
University of Science and Technology of China, Hefei, Anhui 230026, China
**3** Department of Materials Science and Engineering,
University of Washington, Seattle, Washington, 98195, USA
**4** Department of Physics, University of Washington, Seattle, Washington, 98195, USA

⋆ dixiao@uw.edu

## Abstract

We demonstrate that in Weyl semimetals, the momentum-space helical spin texture can couple to the chirality of the Weyl node to generate a frequency-independent optical spin injection. This frequency-independence is rooted in the topology of the Weyl node. Since the helicity and the chirality are always locked for Weyl nodes, the injected spin from a pair of Weyl nodes always add up, implying no symmetry requirements for Weyl semimetals. Finally, we show that such frequency-independent spin injection is robust against multiband corrections and lattice-regularization effect and capable of realizing all-optical magnetization switching in the THz regime.

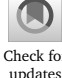

# 1 Introduction

Weyl semimetals, which host topologically protected Weyl nodes with linear dispersion, have been the focus of condensed matter physics in recent years [1]. One of the most celebrated features of Weyl semimetals is that each Weyl node acts as a magnetic monopole in the momentum space, emitting the Berry curvature as the effective magnetic field and possessing a quantized chirality given by the monopole charge. This chiral feature leads to many intriguing experimental consequences [1–8]. In addition to the chirality, the strong spin-orbital coupling in Weyl semimetals also gives rise to the spin-momentum locking around each Weyl node, making the Weyl semimetal a promising candidate in spintronics. For example, it has been proposed theoretically [9] and confirmed experimentally [10, 11] that such spin-momentum locking leads to large spin Hall angle in Weyl semimetals.

In this work, we focus on another feature of the spin-momentum locking, i.e., the helical spin texture around each Weyl node. We show that it plays an essential role in the optical spin injection. The spin injection is a basic tool in opto-spintronics and widely used to generate spin polarized carriers by absorbing light [12, 13]. Especially, the circular optical spin injection further binds the spin polarization of the photoinduced carrier with the circular polarization of light, manifesting as a linearly increasing spin magnetization $M$ that switches with the circular polarization of light [14], i.e.,

$$\frac{dM_i}{dt} = \beta_{ij}(\omega)[i\boldsymbol{E}(\omega) \times \boldsymbol{E}^{\star}(\omega)]_j \,, \tag{1}$$

where $[i\boldsymbol{E}(\omega) \times \boldsymbol{E}^{\star}(\omega)]_j = \pm|\boldsymbol{E}(\omega)|^2$ for right- and left-circularly polarized light. Such circular optical spin injection was mainly studied in bulk semiconductors [12–18] and two-dimensional topological insulators [19, 20], and is generally very sensitive to light frequency.

Interestingly, in Weyl semimetals, the helical spin texture leads to a constant spin injection coefficient, independent of light frequency. We focus on the diagonal part of the coefficient, and show that $\text{Tr}\beta$ is proportional to the product of the chirality and the helicity (see Eq. (4)). It is then found that in addtion to the chirality, the helicity is also a constant around each Weyl node (see Eq. (6)), characterizing the helical spin texture. Moreover, the helicity is proportional to the chirality, and hence the injected spin from a pair of Weyl nodes always add up. Therefore, the injected spin and its frequency-independent feature have no symmetry requirements. Finally, using a tight-binding model of the Weyl semimetal, we demonstrate that such frequency-independence is also robust against the lattice regularization.

An important application of the spin injection is the magnetization reversal by circularly polarized light. The frequency-independence of the injected spin in Weyl semimetals is a highly desirable feature as it eliminates the need to fine tune the light frequency close to resonance. For magnetic Weyl semimetals with realistic experimental setup [21, 22], we estimate that the injected spin can realize optical magnetization switching in the THz regime, highlighting the potential of Weyl semimetals in opto-spintronic applications.

# 2 The circular optical spin injection

We first revisit the general theory of the circular optical spin injection. Under the irradiation of circularly polarized light, a spin injection can be induced acording to Eq. (1). The resulting magnetization grows linearly in time, similar to the current injection in the circular photogalvanic effect (CPGE) [8, 23–26]. It has been proposed previously that the CPGE is frequency-independent and quantized for non-centrosymmetric Weyl semimetals due to the quantized chirality [8]. Here we will show that the topology of the Weyl nodes also leads to

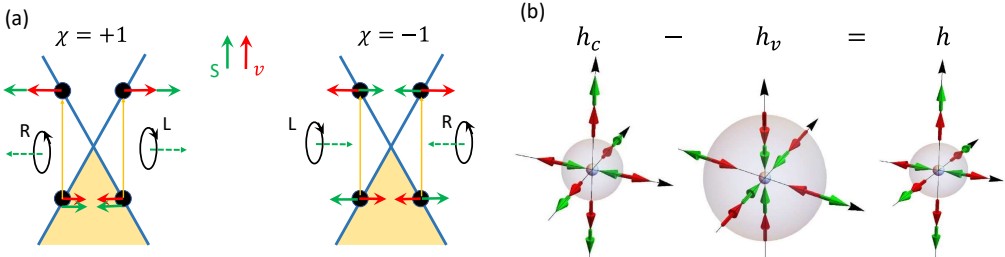

Figure 1: Schematic diagram of the circular optical spin injection for a pair of Weyl nodes connected by inversion symmetry (a) and momentum-space spin texture (b). In (a), the coupling to the circular polarizaiton of light is determined by the Berry curvature in the conduction band, which is parallel (for $\chi = +1$) and antiparallel (for $\chi = -1$) to the velocity $v$. The dashed green arrow in (a) shows the angular momentum of light, which is transferred to the Weyl Fermions, accounting for the difference between the green arrows in the conduction and valence band. When the chirality is flipped from $+1$ to $-1$, the projection of spin on $v$ is also flipped, suggesting a locked helicity and chirality, which collaborate constructively to yield a nontrivial induced magnetization from chiral light. For (b), the three plots show the spin texture for helicity in the conduction ($h_c$) and valence band ($h_v$), and their difference ($h$).

a frequency-independent spin injection coefficient $\beta_{ij}$, albeit with distinct features from the CPGE.

We will focus on the diagonal part of $\beta_{ij}$. Since both $M$ and $E \times E^\star$ transform as axial vectors, $\beta_{ii}$ does not exert any symmetry constraint and hence exists in a wide variety of materials. Specifically, the trace $\mathrm{Tr}\,\beta_{ij}$ simply transforms as a scalar under point group operations. This is in sharp contrast to the CPGE, which requires inversion-symmetry breaking.

The general expression of optical spin injection for any light polarization is derived in Ref. [14], and we also sketch the derivation in Appendix C. By taking the antisymmetrization of the electric field components, we extract the part that is sensitive to the circular polarization of light. The resulting coefficient $\beta_{ii}$ reads (we set $e = \hbar = \mu_B = 1$ hereafter for simplicity)

$$\beta_{ii}(\omega) = \sum_{\ell,n} \int \frac{g_S d\mathbf{k}}{8\pi^2} (\Omega_i)_{n\ell} (\Delta s_i)_{n\ell} \Delta f_{\ell n} \delta(\omega_{\ell n} - \omega), \tag{2}$$

where $g_S$ is the $g$-factor, $(\Omega_i)_{n\ell} = -\epsilon_{ijj'} \mathrm{Im}[\langle u_n | i\partial_{k_j} | u_\ell \rangle \langle u_\ell | i\partial_{k_{j'}} | u_n \rangle]$ is the band-resolved Berry curvature [27], $(\Delta s_i)_{n\ell} = \langle u_n | s_i | u_n \rangle - \langle u_\ell | s_i | u_\ell \rangle$, $\omega_{\ell n} = \varepsilon_\ell - \varepsilon_n$, and $\Delta f_{\ell n} = f_\ell - f_n$ with $f_\ell$ being the Fermi function of band $\ell$. We note that by replacing $g_S(\Delta s)_i$ with the velocity difference $\langle u_n | \hat{v}_i | u_n \rangle - \langle u_\ell | \hat{v}_i | u_\ell \rangle$ in (2), one can obtain the current injection coefficient in the CPGE [8,23].

Equation (2) can be easily understood based on Fermi's golden rule. The coupling between light and matter is described by $\hat{H}' = (i/\omega) v \cdot E$ following the Peierls substitution. Fermi's Golden rule dictates that the oscillator strength of the light absorption has the following feature: $\Gamma_{n \to \ell} \propto |\langle n|\hat{H}'|\ell\rangle|^2$. We then take the antisymmetrization of the electric field components, and obtain the oscillator strength specifically tied to the circular polarization of light [28]: $\Gamma_{n \to \ell}^{cir} \propto (\Omega_i)_{n\ell} i(E \times E^\star)_i$. After the light absorption, the spin will change by $(\Delta s_i)_{n\ell}$. The rate of the change of the spin magnetization is thus given by Eq. (2). Such a process can be heuristically described as in Fig. 1(a).

## 3 Spin injection and the helicity of the Weyl node

The circular optical spin injection in Weyl semimetals is closely tied to the helical spin texture of the Weyl node. To see this, we consider a generic Weyl Hamiltonian as follows

$$\hat{H} = \chi v \boldsymbol{k} \cdot \boldsymbol{\sigma} \,, \tag{3}$$

where $\chi = \pm 1$ is the chirality of the Weyl node, $v$ is the Fermi velocity, and $\boldsymbol{\sigma}$ are the Pauli matrices for the pseudospin indexing the two bands. For such Weyl Hamiltonian, $(\Omega_i)_{n\ell}$ in Eq. (2) reduces to the Berry curvature, which takes a particularly simple form: $\boldsymbol{\Omega} = \chi \boldsymbol{k}/(2k^3)$. Assuming that the Fermi energy falls on the Weyl point, we obtain the following spin injection coefficient

$$
\begin{aligned}
\mathrm{Tr}\beta_{ij} &= g_S \int \frac{d\boldsymbol{k}}{8\pi^2} \frac{\chi v^3}{2\varepsilon^3} (\boldsymbol{k} \cdot \Delta \boldsymbol{s}) \delta(2\varepsilon - \omega) \\
&= \frac{g_S}{8\pi v} \chi h(\omega),
\end{aligned}
\tag{4}
$$

where $h(\omega)$ is given by

$$h(\omega) = \oint_{2\varepsilon=\omega} \frac{1}{g(\varepsilon)} \frac{\boldsymbol{v}}{v} \cdot \Delta \boldsymbol{s} \frac{d\boldsymbol{k}}{8\pi^3} \,. \tag{5}$$

Here $\boldsymbol{v}$ is the velocity in the conduction band with $\varepsilon$ being the band energy, $g(\varepsilon) = \varepsilon^2/(2\pi^2 v^3)$ is the density of states at $\varepsilon$, and $\Delta \boldsymbol{s} = \langle u_c|\boldsymbol{s}|u_c\rangle - \langle u_v|\boldsymbol{s}|u_v\rangle$ is the $\boldsymbol{k}$-resolved spin difference between the conduction ($c$) and the valence ($v$) band.

The factor $h(\omega)$ in Eq. (4) is the helicity of the Weyl node. From Eq. (5), we find that $h(\omega)$ contains the projection of the spin difference onto the band velocity. Since the band velocity is porportional to $\boldsymbol{k}$, $h(\omega)$ is then the projection of the spin on momentum, i.e., it is the helicity, following the definition in particle physics. Note that the helicity $h(\omega)$ here is different from that in two-dimensional systems [29, 30].

Importantly, we find that $h(\omega)$ is an inherent property of the Weyl node, independent of the frequency, which eventually leads to a frequency-independent $\beta_{ii}$. To start with, we consider the simplest scenario where $\boldsymbol{\sigma}$ in the Weyl Hamiltonian operates in the real spin space, and has a basis of $|\uparrow\rangle$ and $|\downarrow\rangle$, as proposed for the Kramers-Weyl Hamiltonian [31]. It is straightforward to show that the helicity coincides with the chirality: $h(\omega) = \chi$, i.e. it is frequency-independent.

Generally speaking, $\boldsymbol{\sigma}$ operates in a mixed spin-orbital space. Its basis, labeled by $|+\rangle$ and $|-\rangle$, is determined by projecting the full crystal Hamiltonian with the spin-orbital coupling onto the two-band Weyl Hamiltonian. In this case, the helicity reads (see Appendix A)

$$h(\omega) = \frac{\chi}{3}[\langle +|s_z|+\rangle - \langle -|s_z|-\rangle + (\langle +|s_+|-\rangle + \mathrm{c.c.})] \overset{\mathrm{def}}{=} h_0 \,, \tag{6}$$

where $s_+ = s_x + is_y$. One immediately finds that the frequency-independence is preserved.

The helicity quantifies the spin texture near a Weyl node: it varies in the range $(-\chi/3, \chi)$ for different textures (see Appendix. B). We find that the helicity is generally nonzero, unless under stringent algebraic conditions. To illustrate the helicity, we define the conduction and valence band helicity $h_\alpha$ by replacing $\Delta \boldsymbol{s}$ in Eq. (5) with $\langle u_\alpha|\boldsymbol{s}|u_\alpha\rangle$, where $\alpha = c, v$ for the conduction and valence band, respectively. Then $h$ can be written as $h = h_c - h_v$. An example of the spin texture around a Weyl node with a nonzero helicity is shown in Fig. 1(b) (this spin texture is obtained using the effective model in Sect. 4). We also note that by identifying the basis $|+\rangle$ and $|-\rangle$ as different spin states, the pattern of the spin-momentum locking changes,

Table 1: Basis with different spin flavor, the Weyl Hamiltonian in the spin space, the helicity, and the spin injection coefficient.

| Basis | Hamiltonian in the spin space | $h_0$ | $\text{Tr}\beta_{ij}$ |
|---|---|---|---|
| $\|+\rangle = \|\uparrow\rangle$ <br> $\|-\rangle = \|\downarrow\rangle$ | $\hat{H} = vk_x s_x + vk_y s_y + vk_z s_z$ [1] | $1$ | $\frac{g_S}{8\pi v}$ |
| $\|+\rangle = e^{i\pi/4}\|\uparrow\rangle$ <br> $\|-\rangle = e^{-i\pi/4}\|\downarrow\rangle$ | $\hat{H} = vk_y s_x - vk_x s_y + vk_z s_z$ [2] | $\frac{1}{3}$ | $\frac{g_S}{24\pi v}$ |
| $\|+\rangle = \frac{1}{2}\|\uparrow\rangle + \frac{\sqrt{3}}{2}i\|\downarrow\rangle$ <br> $\|-\rangle = \frac{\sqrt{3}}{2}i\|\uparrow\rangle + \frac{1}{2}\|\downarrow\rangle$ | $\hat{H} = vk_x s_x - v\left(\frac{1}{2}k_y - \frac{\sqrt{3}}{2}k_z\right)s_y - v\left(\frac{1}{2}k_z + \frac{\sqrt{3}}{2}k_y\right)s_z$ | $0$ | $0$ |
| $\|+\rangle = -i\|\downarrow\rangle$ <br> $\|-\rangle = i\|\uparrow\rangle$ | $\hat{H} = -vk_x s_x - vk_y s_y - vk_z s_z$ | $-\frac{1}{3}$ | $-\frac{g_S}{24\pi v}$ |

[1] The Kramers-Weyl Hamiltonian [31].
[2] The three-dimensional Rashba spin-orbital coupling [30].

which can also manifest as different Weyl Hamiltonians in the real spin space. In Table. 1, we show several examples of the basis, with the corresponding average helicity and the spin injection coefficient.

From Eq. (6), we see that $h_0$ is always proportional to $\chi$. As a result, the spin injection coefficient $\text{Tr}\beta$ is quadratic in $\chi$ [cf. Eq. (4)]. Therefore, contributions from a pair of Weyl nodes with opposite chiralities always add up, as illustrated in Fig. 1(a). This is in sharp contrast to the injection current, which is linear in $\chi$ [8]. This microscopic difference manifests as their different requirements of symmetries: the injection current requires the Weyl semimetals to possess structural chirality [8], while the spin injection does not.

If there are $N > 1$ pairs of Weyl nodes, the resulting frequency-independent spin injection is simply the summation of that for each pair of Weyl nodes. If different pairs of Weyl nodes are further connected by symmetries, the signal is simply $N$ times contribution from a single pair, as the diagonal part of the spin injection is insensitive to any symmetry operation.

When the Fermi energy $\mu$ is away from the Weyl point, optical transition is forbidden for light frequency lower than $2|\mu|$ and hence $\text{Tr}\beta$ vanishes. The frequency-independent spin injection is recovered once the frequency is above $2|\mu|$.

In Weyl semimetals, the Weyl node may be anisotropic, due to either different Fermi velocities along different directions or tilting. For the former case, the frequency-independence still persists, but the expression for $\beta_{ii}$ is slightly modified (see Appendix. A). For the latter case, the frequency independence is perserved as long as the Weyl node is not of type-II.

## 4 Lattice model

The above discussion is for perfect Weyl nodes. However, the Weyl cone is generally subject to lattice regularization, leading to the deviation from linear dispersion as well as multiband structures. To test their effects on the frequency-independence of the spin injection, we consider a Weyl semimetal regularized on a cubic lattice, with the Hamiltonian given by [32]

$$\hat{H} = \lambda \tau_x^o(\tau_x^s \sin k_x + \tau_y^s \sin k_y + \tau_z^s \sin k_z) + \epsilon \tau_z^o + b\tau_z^s, \tag{7}$$

where the Pauli matrices $\boldsymbol{\tau}^o$ and $\boldsymbol{\tau}^s$ operate in orbital and real-spin space, with the basis given by $(|A\rangle, |B\rangle)$ and $(|\uparrow\rangle, |\downarrow\rangle)$, respectively, $\epsilon = m + r(3 - \cos k_x - \cos k_y - \cos k_z)$, $b$ is the Zeeman field, $\lambda$ is the strength of the spin-orbital coupling, and $\boldsymbol{k}$ is the dimensionless momentum (i.e.,

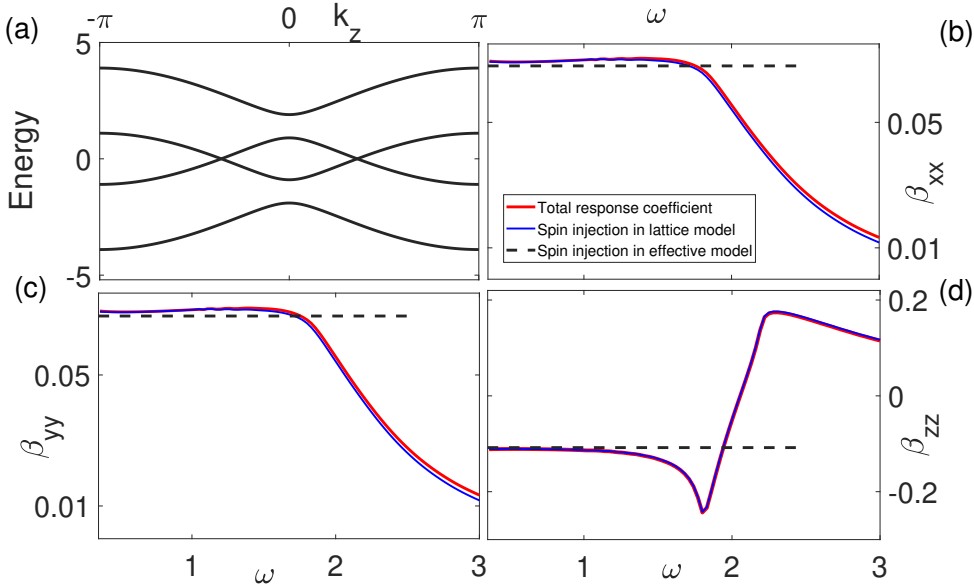

Figure 2: The circular optical spin injection coefficient in a tight-binding model. Panel (a) is the spectrum showing two Weyl nodes. Panel (b)-(d) contain the spin injection for the effective two-band Weyl Hamiltonian (black lines), the spin injection for the full lattice model (blue lines), and the total spin magnetization (red lines). Energy in (a) and frequency in (b-d) are in units of the model parameter $r$. $\beta$ is in units of $e^2 \tau_0 \mu_B / (2\hbar r a)$.

we set the lattice constant $a = 1$). Such a model can host two Weyl nodes located on the $k_z$ axis, as shown in Fig. 2(a).

By projecting the four-band Hamiltonian onto the Weyl cone, we find that the basis indeed possesses a nontrivial spin flavor, inherited from the spin-orbit coupling (first term in $\hat{H}$) and the Zeeman coupling (last term in $\hat{H}$). Specifically, for the Weyl node on the positive $k_z$ axis, the Weyl Hamiltonian reads $\hat{H} = v_x k_x \sigma_x - v_y k_y \sigma_y + v_z k_z \sigma_z$ and the basis for $\boldsymbol{\sigma}$ reads

$$
\begin{aligned}
|+\rangle &= 0.343|A\uparrow\rangle - 0.939|B\uparrow\rangle, \\
|-\rangle &= -0.939|A\downarrow\rangle + 0.343|B\downarrow\rangle.
\end{aligned}
\tag{8}
$$

Such a spin structure indeed endows a nonzero helicity $h(\omega) = 0.1$ for this Weyl node.

We then calculate $\beta_{ii}$ within the effective two-band $\boldsymbol{k} \cdot \boldsymbol{p}$ model as well as the full four-band lattice model. As shown in Fig. 2(b)-(d), up to about $\omega = 1.7r$, the latter (blue lines) only deviates slightly from the former (dotted black lines), and both of them are nearly flat, suggesting a robust frequency-independence feature against lattice regularization. When the light frequency is close to $2r$, the optical transition occurs in the region where the two Weyl nodes start to intercept each other, and the frequency-independence is gradually destroyed by the lattice effect.

Finally, we comment that in addition to the spin injection, the photoinduced spin magnetization also contains a static part, similar to the static photocurrent in the circular photogalvanic effect [24–26]. Using the density-matrix perturbation theory one can obtain the full expression for the photo-induced spin magnetizaiton (see Appendix. C), which has the form: $\beta_{ij}^{tot} = \tau_0 \beta_{ij} + \gamma_{ij}$ with $\tau_0$ being the relaxation time and $\gamma_{ij}$ being the contribution from the static photoinduced spin magnetization. However, using the previous tight-binding model, we find that the spin injection generally dominates over the static spin magnetization. Compared to the inherent energy scale of the electronic structure, which is on the order of eV, the

relaxation process generally involves a much smaller energy scale. To reflect this, in the tight-binding model, we take $1/\tau_0 = 0.02r$, and calculate $\tau_0\beta_{ii}$ as well as the $\beta_{ii}^{\text{tot}}$ in Eq. (C.11). As shown in Fig. 2 (b)-(d), the latter (red lines) almost coincide with the former (blue lines) over the whole frequency range, confirming the dominance of the injection contribution to the spin magnetization. As a result, the total photoinduced magnetization approximately preserves the frequency-indepent feature up to $\omega = 1.7r$.

# 5 THz optical switching

In practice, the photoinduced spin magnetization can be used for swithcing the inherent magnetic order of ferromagnetic materials [12, 13, 21, 33, 34]. For this purpose, the frequency-independence is a valuable feature: usually the photoinduced magnetization is very sensitive to frequency, and becomes strong only close to resonance. The frequency-independence over a wide range can thus remove the need to find monochrome light with fine tuned frequency.

Magnetic Weyl semimetal phase can exist in several materials [35–39]. Their magnetic orders can then be manipulated by the large spin injection in the THz regime. We assume a laser fluence 0.2 mJ/cm$^2$ with a duration of 100 fs [21, 22], a lattice constant 2 Å, and a spin injection coefficient $10^{-10}$ V$^{-2}$m$^{-1}$J. This translates to a photoinduced spin magnetization around $0.025\mu_B$ per unit cell. This induced magnetization affects the inherent magnetic order through the following exchange coupling

$$\hat{H} = J\boldsymbol{S}_{loc} \cdot \boldsymbol{s}_{el}, \tag{9}$$

where $J$ is the exchange coupling strength, $\boldsymbol{S}_{loc}$ is the local angular momentum, and $\boldsymbol{s}_{el}$ is the electronic angular momentum. Based on this coupling, the induced magnetization amounts to an effective field $\boldsymbol{B}_{eff} = J\boldsymbol{m}_{el}/(2\mu_B)$ for the switching of $\boldsymbol{S}_{loc}$ through a field-like or damping-like torque. Assuming a typical exchange coupling strength of 150 meV, we find that the angular frequency of the magnetization precession is $J|\boldsymbol{m}_{el}|/(2\mu_B\hbar) = 3$ THz.

# 6 Conclusion

In conclusion, we demonstrate that the spin-momentum locking in Weyl semimetals yields a helical spin texture around each Weyl node, and recognize the helicity as an inherent property that quantifies such spin texture. We show that the circular optical spin injection in Weyl semimetals is proportional to the helicity, and hence frequency-independent. Such frequency-independence is robust against lattice effect and dominates over other contributions to the photoinduced spin magnetization. The spin injection is also large enough to realize all-optical magnetization switching in the THz regime.

## Acknowledgments

We acknowledge insightful discussions with Dazhi Hou and Xiaodong Xu.

**Funding information** Y.G. acknowledge support from the Fundamental Research Funds for the Central Universities, China (Grant No. WK2340000102). Work at the University of Washington is supported by the Department of Energy, Basic Energy Sciences, Grant No. DE-SC0012509.

# A   Evaluation of the helicity and spin injection for the Weyl Hamiltonian

To evaluate the average helicity, we note that the basis for the Weyl Hamiltonian contains spin flavor. We can then project the spin operator on the basis of $\boldsymbol{\sigma}$:

$$s = \begin{pmatrix} \langle +|\boldsymbol{s}|+\rangle & \langle +|\boldsymbol{s}|-\rangle \\ \langle -|\boldsymbol{s}|+\rangle & \langle -|\boldsymbol{s}|-\rangle \end{pmatrix}. \tag{A.1}$$

The eigen wave function in the conduction and valence band reads

$$\psi_c = \frac{1}{\xi}(a,b)^{\mathrm{T}}, \qquad \psi_v = \frac{1}{\xi}(-b,a^\star)^{\mathrm{T}}, \tag{A.2}$$

where $a = \chi v(k_x - ik_y)$, $b = \varepsilon - \chi v k_z$, $\varepsilon = vk$ and $\xi = \sqrt{2\varepsilon(\varepsilon - \chi v k_z)}$.

With the help of the analytic expression of the wave function, we can obtain the spin expectation value in conduction and valence band, i.e.,

$$(s_i)_c = \frac{1}{\xi^2}[|a|^2\langle +|s_i|+\rangle + |b|^2\langle -|s_i|-\rangle + (a^\star b\langle +|s_i|-\rangle + c.c.)],$$
$$(s_i)_v = \frac{1}{\xi^2}[|a|^2\langle -|s_i|-\rangle + |b|^2\langle +|s_i|+\rangle - (ba^\star\langle +|s_i|-\rangle + c.c.)]. \tag{A.3}$$

The spin difference is given by

$$\begin{aligned} \Delta s_i &= (s_i)_c - (s_i)_v \\ &= \frac{2\varepsilon^2}{v^2}\Omega_z(\langle +|s_i|+\rangle - \langle -|s_i|-\rangle) + \frac{2\varepsilon^2}{v^2}[(\Omega_x + i\Omega_y)\langle +|s_i|-\rangle + c.c.]. \end{aligned} \tag{A.4}$$

The average helicity can then be evaluated:

$$\begin{aligned} h &= \frac{v^2}{4\pi\varepsilon_0^2}\int \boldsymbol{v}\cdot\Delta\boldsymbol{s}\,\delta(\varepsilon - \varepsilon_0)d\boldsymbol{k} \\ &= \frac{\chi}{3}[\langle +|s_z|+\rangle - \langle -|s_z|-\rangle + (\langle +|s_+|-\rangle + c.c.)]. \end{aligned} \tag{A.5}$$

For an anisotropic Weyl node the above procedure still works but some of the expressions are slightly modified. For example, the $z$-component of the Berry curvature now reads $\Omega_z = \chi v_x v_y v_z k_z/(2\varepsilon^3)$ with $\varepsilon = \sqrt{v_x^2 k_x^2 + v_y^2 k_y^2 + v_z^2 k_z^2}$. Then the $zz$-component of the spin injection coefficient reads

$$\beta_{zz} = g_S\int \frac{d\boldsymbol{k}}{8\pi^2}\frac{\chi v_x v_y v_z k_z}{2\varepsilon^3}(\Delta s)_z\delta(2\varepsilon - \omega). \tag{A.6}$$

The eigen wave function in the conduction and valence band is still given by Eq. (A.2), but with $a = \chi(v_x k_x - iv_y k_y)$ and $b = \varepsilon - \chi v_z k_z$. The $z$-th component of the spin difference then has the following form

$$\Delta s_i = \frac{\chi v_z k_z}{\varepsilon}(\langle +|s_i|+\rangle - \langle -|s_i|-\rangle) + \frac{\chi(v_x k_x + iv_y k_y)\langle +|s_i|-\rangle + c.c.}{\varepsilon}. \tag{A.7}$$

Therefore,

$$\begin{aligned} \beta_{zz} &= g_S\int \frac{d\boldsymbol{k}}{8\pi^2}\frac{v_x v_y v_z^2 k_z^2}{2\varepsilon^4}(\langle +|s_z|+\rangle - \langle -|s_z|-\rangle)\delta(2\varepsilon - \omega) \\ &= \frac{g_S}{24\pi v_z}(\langle +|s_z|+\rangle - \langle -|s_z|-\rangle). \end{aligned} \tag{A.8}$$

Similarly, we can obtain the $xx$ and $yy$ component of the spin injection coefficient as follows

$$\begin{aligned}
\beta_{xx} &= \frac{g_S}{24\pi v_x}(\langle+|s_x|-\rangle + \text{c.c.}), \\
\beta_{yy} &= \frac{g_S}{24\pi v_y}(\langle+|is_y|-\rangle + \text{c.c.}).
\end{aligned} \tag{A.9}$$

## B  Range of the average helicity

In this section, we derive the condition for $h = 0$. We note that $h$ can be reformulated as $h = \chi \langle A\rangle/3$ with

$$\langle A\rangle = \begin{pmatrix} \langle+| & \langle-| \end{pmatrix} \begin{pmatrix} s_z & s_x + is_y \\ s_x - is_y & -s_z \end{pmatrix} \begin{pmatrix} |+\rangle \\ |-\rangle \end{pmatrix}. \tag{B.1}$$

To see when $\langle A\rangle = 0$, we first consider the eigenvalue problem:

$$\begin{pmatrix} s_z & s_x + is_y \\ s_x - is_y & -s_z \end{pmatrix} \begin{pmatrix} \psi_a \\ \psi_b \end{pmatrix} = \lambda \begin{pmatrix} \psi_a \\ \psi_b \end{pmatrix}. \tag{B.2}$$

We then have

$$\begin{aligned}
s_+\psi_b &= (\lambda - s_z)\psi_a, \\
s_-\psi_a &= (\lambda + s_z)\psi_b.
\end{aligned} \tag{B.3}$$

If $\lambda \neq 1/2$, we have

$$(\lambda - s_z)^{-1}s_+\psi_b = \psi_a. \tag{B.4}$$

Then we have

$$s_-(\lambda - s_z)^{-1}s_+\psi_b = (\lambda + s_z)\psi_b. \tag{B.5}$$

In the basis of spin-z states ($|\uparrow\rangle, |\downarrow\rangle$), we can write $\psi_b = (b_1, b_2)^T$ and $\psi_a = (a_1, a_2)^T$. We will further label $\psi = (a_1, a_2, b_1, b_2)^T$. Then we get two solutions for the equations of $\psi_b$. The first one is

$$a_2 = b_1 = 0, \qquad \lambda = \frac{3}{2}. \tag{B.6}$$

The second one is

$$b_2 = -a_1, \qquad \lambda = -\frac{1}{2}. \tag{B.7}$$

Therefore, $\lambda$ has two different values. For $\lambda = 3/2$, the corresponding helicity is $h = \chi$, and we have only one eigenstate

$$|\psi\rangle_1 = (1, 0, 0, 1)^T. \tag{B.8}$$

In this case, $\psi_a\rangle = |\uparrow\rangle$ and $|\psi_b\rangle = |\downarrow\rangle$. Therefore, the Pauli matrices $\boldsymbol{\sigma}$ stands for real spin and the resulting Weyl Hamiltonian describes the Kramers-Weyl node.

For $\lambda = -1/2$, the corresponding helicity is $h = -\chi/3$ and we have three different eigenstates

$$\begin{aligned}
|\psi\rangle_2 &= (1, 0, 0, -1)^T, \\
|\psi\rangle_3 &= (0, 0, 1, 0)^T, \\
|\psi\rangle_4 &= (0, 1, 0, 0)^T.
\end{aligned} \tag{B.9}$$

We will first discuss the solution $|\psi\rangle_2$. In this case, $|\psi\rangle_a = |\uparrow\rangle$ and $|\psi\rangle_b = -|\downarrow\rangle$. The spin Pauli matrices $s$ is related to $\sigma$ by: $s_x = -\sigma_x$, $s_y = -\sigma_y$, and $s_z = \sigma_z$. The corresponding Weyl Hamiltonian reads: $\hat{H} = v(\mathbf{k} \cdot \boldsymbol{\sigma}) = v(-k_x s_x - k_y s_y + k_z s_z)$.

However, the other two eigenstates $|\psi\rangle_3$ and $|\psi\rangle_4$ are not physical solutions, due to one of $|\psi\rangle_a$ and $|\psi\rangle_b$ are zero. But their combination is generally physical. To see this, we expand the basis of the Weyl cone with the above eigenstates

$$(|+\rangle, |-\rangle)^T = \sum_i c_i |\psi\rangle_i = (c_1 + c2, c_4, c_3, c_1 - c_2)^T. \tag{B.10}$$

Such a state is physical as long as both $|+\rangle$ and $|-\rangle$ are nonzero, i.e. (1) $c_4$ and $c_1 + c_2$ cannot all be zero; (2) $c_3$ and $c_1 - c_2$ cannot all be zero. Moreover, both $|+\rangle$ and $|-\rangle$ should be orthonormalized, requiring (3) $|c_1 + c_2|^2 + |c_4|^2 = 1$, (4) $|c_3|^2 + |c_1 - c_2|^2 = 1$, and (5) $(c_1 + c_2)^\star c_3 + c_4^\star(c_1 - c_2) = 0$.

For physical states, $\langle A \rangle = 0$ if and only if

$$3|c_1|^2 - |c_2|^2 - |c_3|^2 - |c_4|^2 = 0. \tag{B.11}$$

If the Weyl cone does not reside at any high-symmetry point, this condition is only met accidentally. We can safely say that generally $\langle A \rangle \neq 0$ and hence $h \neq 0$.

Moreover, the two values of $\lambda$ gives the maximum values of $h$, as any combination of eigenstates will always takes values between them. Therefore, we know that $-\chi/3 \leq h \leq \chi$.

## C Full expression for the spin injection

In this section, we first sketch the derivation of the full contribution to the photoinduced spin magnetization using the density-matrix perturbation theory. Under the light irradation, the Hamiltonian is modified according to the Peierls substitubtion: $\hat{H}(\mathbf{p}) \to \hat{H}(\mathbf{p} + e\mathbf{A})$. Here we adopt the velocity gauge for the light electric field, which is equivalent to the length gauge. We then expand the Hamiltonian $\hat{H} = \hat{H}_0 + \hat{H}'$, where $\hat{H}_0$ is the unperturbed Hamiltonian and $\hat{H}'$ is the perturbation which has the following form

$$\hat{H}' = \sum_{i=1}^2 \mathbf{A}(\omega_i) \cdot \hat{\mathbf{v}} e^{-i\omega_i t}, \tag{C.1}$$

where $\mathbf{A}(\omega)$ is the vector potential from light, $\hat{\mathbf{v}}$ is the velocity operator. Similar to Ref. [23], we first set $\omega_1 + \omega_2 = \Omega$ and will set $\Omega \to 0$ in the end. We assume that the perturbation is switched adiabatically. In other words, the frequency should have an imaginary part: $\pm\omega \to \pm\omega + i\eta$. This imaginary part is always positive due to the requirement of causality. Therefore, $\Omega \to \Omega + i/\tau_0$ with $\tau_0 = 1/(2\eta)$.

We have ignored the second order perturbation because it does not respond to the circular polarization of light. To see this, note that the second order perturbation generally has the following form

$$\hat{H}'' = \hat{\Gamma}_{ij} A_i(\omega_k) A_j(\omega_\ell) e^{-i(\omega_k + \omega_\ell)t}, \tag{C.2}$$

where $\hat{\Gamma}_{ij}$ is the Hessian operator, which is symmetric with respect to $i$ and $j$. Here and hereafter, repeated indices are always summed unless otherwise specified. However, the circular polarization requires the antisymmetrization of $i$ and $j$. As a result, $\hat{H}''$ does not respond to the circular polarization of light.

With the perturbation from light electric field, the density operator $\hat{\rho}$ evolves according to the following equation of motion

$$\frac{d\hat{\rho}}{dt} = -i[\hat{H}, \hat{\rho}]. \tag{C.3}$$

We expand the density operator with respect to the vector potential: $\rho = \rho_0 + \delta\rho^{(1)} + \delta\rho^{(2)} + \cdots$ At the first order, we have

$$\frac{d\delta\rho^{(1)}}{dt} = -i[\hat{H}_0, \delta\rho^{(1)}] - i[\hat{H}', \rho_0]. \tag{C.4}$$

The solution reads

$$\delta\rho^{(1)} = -iA_j(\omega_i)e^{-i\omega_i t}\int_0^\infty d\tau\, e^{-i\hat{H}_0\tau}[\hat{v}_j, \hat{\rho}_0]e^{i\hat{H}_0\tau}e^{i\omega_i\tau}, \tag{C.5}$$

where $\hat{\rho}_0$ is the density operator for the unperturbed system with the Hamiltonian $\hat{H}_0$. At second order, we have

$$\frac{d\delta\rho^{(2)}}{dt} = -i[\hat{H}_0, \delta\rho^{(2)}] - i[\hat{H}', \rho^{(1)}]. \tag{C.6}$$

The solution reads

$$\delta\rho^{(2)} = -A_k(\omega_i)A_\ell(\omega_j)e^{-i(\omega_i+\omega_j)t}\int_0^\infty dt_1 dt_2\, e^{-i\hat{H}_0 t_1}[\hat{v}_k,[\hat{v}_\ell(-t_2),\hat{\rho}_0]]e^{i\hat{H}_0 t_1}e^{i(\omega_i+\omega_j)t_1}e^{i\omega_j t_2}, \tag{C.7}$$

where $\hat{v}_k(t) = e^{i\hat{H}_0 t}\hat{v}_k e^{-i\hat{H}_0 t}$.

The spin magnetization is obtained as follows: $M_a = \text{Tr}(g_S \hat{s}_a \rho)$. At second order, we have

$$M_a = g_S \text{Tr}(\delta\rho^{(2)}\hat{s}_a). \tag{C.8}$$

Here $g_S$ is the g-factor for spin. We will keep terms that oscillates at the frequency $\Omega$.

An explicit expression can be obtained by inserting the resolution of identity appropriately. For example, we consider the following term in $M_a$ (we use the shorthand $|n\rangle$ for $|u_n\rangle$)

$$-g_S A_k(\omega_i)A_\ell(\omega_j)e^{-i(\omega_i+\omega_j)t}\text{Tr}\int_0^\infty dt_1 dt_2\, \hat{s}_a e^{-i\hat{H}_0 t_1}\hat{v}_k\hat{v}_\ell(-t_2)\hat{\rho}_0 e^{i\hat{H}_0 t_1}e^{i(\omega_i+\omega_j)t_1}e^{i\omega_j t_2}$$

$$= -g_S A_k(\omega_i)A_\ell(\omega_j)e^{-i(\omega_i+\omega_j)t}\int_0^\infty dt_1 dt_2\langle n_1|\hat{s}_a|n_2\rangle\langle n_2|e^{-i\hat{H}_0 t_1}\hat{v}_k|n_3\rangle$$

$$\times \langle n_3|\hat{v}_\ell(-t_2)\hat{\rho}_0 e^{i\hat{H}_0\tau_1}|n_1\rangle e^{i(\omega_i+\omega_j)t_1}e^{i\omega_j t_2}$$

$$= -g_S \sum_{i\neq j}A_k(\omega_i)A_\ell(\omega_j)e^{-i\Omega t}\int_0^\infty dt_1 dt_2 (s_a)_{n_1 n_2}(v_k)_{n_2 n_3}(v_\ell)_{n_3 n_1}f_{n_1}e^{i(\Omega+\omega_{n_1 n_2})t_1}e^{i(\omega_j+\omega_{n_1 n_3})t_2}$$

$$= g_S \sum_{i\neq j}A_k(\omega_i)A_\ell(\omega_j)e^{-i\Omega t}(s_a)_{n_1 n_2}(v_k)_{n_2 n_3}(v_\ell)_{n_3 n_1}f_{n_1}\frac{1}{\Omega+\omega_{n_1 n_2}}\frac{1}{\omega_j+\omega_{n_1 n_3}}, \tag{C.9}$$

where $\omega_{n_1 n_3} = \varepsilon_{n_1} - \varepsilon_{n_3}$. Here and hereafter, the integration over the Brillouin zone is temporarily dropped unless otherwise specified.

Using the similar treatment for the remaining terms in $M_a$, finally we have

$$M_a = g_S \sum_{i\neq j}A_k(\omega_i)A_\ell(\omega_j)e^{-i\Omega t}(s_a)_{n_1 n_2}(v_k)_{n_2 n_3}(v_\ell)_{n_3 n_1}\frac{1}{\Omega+\omega_{n_1 n_2}}\frac{f_{n_1}-f_{n_3}}{\omega_j+\omega_{n_1 n_3}}$$

$$- g_S \sum_{i\neq j}A_k(\omega_i)A_\ell(\omega_j)e^{-i\Omega t}(s_a)_{n_1 n_2}(v_\ell)_{n_2 n_3}(v_k)_{n_3 n_1}\frac{1}{\Omega+\omega_{n_1 n_2}}\frac{f_{n_3}-f_{n_2}}{\omega_j+\omega_{n_3 n_2}}. \tag{C.10}$$

From this, we can define a coefficient $\gamma_{ij}$: $M_i = \beta_{ij}^{\text{tot}}[i\boldsymbol{E}(\omega)\times\boldsymbol{E}^\star(\omega)]_j$. The diagonal part of $\beta_{ij}^{\text{tot}}$ reads

$$\beta_{ii}^{\text{tot}} = \frac{-ig_S\epsilon_{ijk}}{\omega^2}\sum_{\ell,m,n}\int\frac{d\boldsymbol{k}}{16\pi^3}\frac{(v_j)_{m\ell}(v_k)_{\ell n}}{\omega_{nm}+i/\tau_0}\times(G_{\ell n}+G_{m\ell})(s_z)_{nm}, \tag{C.11}$$

where $(v_i)_{m\ell}$ and $(s_z)_{nm}$ are the velocity and spin matrix element, respectively, $\tau_0$ is the relaxation time, and $G_{\ell n} = \Delta f_{\ell n}/(\omega_{\ell n} - \omega - i/\tau_0) - (\omega \to -\omega)$.

Equation. (C.11) yields the spin injection at saturation by taking $n = m$: $\beta_{ii}^{\text{tot}}|_{n=m} = \tau_0 \beta_{ii}$. This is similar to the saturated current from the injection current in the CPGE. Term with $n \neq m$ corresponds to a static photoinduced spin magnetization and is generally nonzero.

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
