# Peer review of "Frequency-independent Optical Spin Injection in Weyl Semimetals"

_SciPost Physics Core, doi:SciPost Phys. Core 7, 002 (2024)_

## Round 1 · Referee Report · Anonymous (Referee 1) · 2023-1-29

Strengths

1 - Addresses a problem (optical spin injection in Weyl semimetals) that, to my knowledge, had not been previously investigated in the literature.

2 - Finds a noteworthy feature in that problem (frequency independence, under certain assumptions) that may stimulate further investigations, and articulates well why such a feature is desirable.

3 - Clearly written, well-organized, good balance between the material in the main text and in the appendices.

4 - Numerical results illustrate in a useful way the main ideas.

5 - Provides estimates in Sec. 5 that could serve as a guide for future experimental efforts.

Weaknesses

1 - As stated in Sec. 2 of the manuscript, the main idea behind this work is partly inspired by Ref. [8], where the current injection in the circular photogalvanic effect was found to be frequency-independent and quantized (under certain assumptions) in acentric Weyl semimetals, only depending on fundamental constants. The corresponding result found in the present work for spin injection is somewhat weaker, lacking the quantization aspect that was the most salient feature of the work presented in Ref. [8]. Specifically, the helicity of a Weyl node defined by Eqs. (5,6) is not quantized, as illustrated in Table 1. In that sense, the main finding of this work is less "topological" than that of Ref. [8] (which is also not truly topological, as it relies on the 2-band approximation).

2 - While the specific problem addressed in this work may not have been previously addressed in the literature, several theoretical works have looked at related problems in Weyl and/or Dirac semimetals. For example,

http://dx.doi.org/10.1103/PhysRevB.93.201202
https://doi.org/10.1103/PhysRevB.101.174429
https://arxiv.org/abs/2009.01388v1
https://doi.org/10.1103/PhysRevLett.126.247202

It would be useful to refer to such works in the Introduction or Conclusion[s], to help place the present work in the proper context.

Report

Of the acceptance criteria for SciPost Physics indicated in

https://scipost.org/SciPostPhys/about#criteria

my assessment is that this work does not quite meet criteria 1, 2, and 4. As for criterion 3,

"Open a new pathway in an existing or a new research direction, with clear potential for multipronged follow-up work."

I believe that it is at partly satisfied. However, my understanding is that it has been quite challenging to identify materials that display in a good approximation the idealized Weyl-semimetal behavior, without it being masked by other "trivial" bands crossing the Fermi level. It is therefore unclear to what extent the spin-injection behavior of a real "Weyl semimetal" would be dominated by the mechanism discussed in the manuscript. This also casts some doubts on the reliability of the estimates for realizing all-optical magnetization switching in the THz range.

Overall, I am unsure as to whether this submission meets the criteria of SciPost Physics, but I am confident that it does meet those of SciPost Physics Core, where it could be published.

Requested changes

1 - In Fig. 1(b), shouldn't the green (spin) arrows in $h_c$ all be pointing outwards, and those in $h_v$ all be pointing inwards ("hedgehog" pattern)?

2 - The caption of Fig. 1 refers to the "chirality" of light. Shouldn't it be "helicity", since it is referenced to the propagation direction?

3 - Figures 2 and 3 are almost identical, the only difference being that Fig. 3 contains one additional curve in panels (b,c,d). Would it make sense to simply replace Fig. 2 with Fig. 3, and change the text accordingly? (Possibly expand the last paragraph of Sec. 4 to include some of the material in the last paragraph of Appendix C).

4 - In both Figs. 2 and 3, the frequency range in panel (c) goes exactly from 0 to 3, whereas in panels (b,c) it goes slightly beyond 3, so that panels (b,d) are slightly "misaligned". Once they are aligned the tick labels above panel (b) could be removed, making the figure less busy.

5 - Equation (10) is the same as (4), with an extra equality in the middle. Maybe replace Eq. (4) with (10), reducing by one the total number of equations?

6 - In Appendix C, $\tau_1$ and $\tau_2$ are dummy time-integration variables, while $\tau_0$ is the relaxation time. Would it be more clear to rename $\tau_1$ and $\tau_2$ as $t_1$ and $t_2$?

7 - The relaxation time $\tau_0$ appears out of nowhere in Eq. (41). The way it is usually introduced is in the adiabatic switching on of the coupling to light. But do I understand correctly that then it should always appear in the combination $\omega+i/\tau_0$? That does not seem to be the case in the denominator of Eq. (41). Also, in the definition of $G_{ln}$ below Eq. (41) is it really just $\omega\rightarrow -\omega$, or should the sign change affect $i/\tau_0$ as well?

8 - Slightly inconsistent notation throughout the text concerning the traced quantity. It is variously written as $\text{Tr} \beta^\text{inj}_{ij}$ [Eq. (4)], $\text{Tr} \beta_{ij}$ [Eq. (10) and in the text above Eq. (2)], and $\text{Tr} \beta^\text{inj}$ (below Table 1). Likewise, it is written $\beta_{zz}$ in Eqs. (16,18), but $\beta_{xx}^\text{inj}$ and $\beta_{yy}^\text{inj}$ in Eq. (19).

9 - The title refers to "spin injection", but the abstract talks about "injection spin", and both forms are used in the main text. Use consistently the first form?

10 - In the abstract, "multiband and lattice effect" should be replaced for clarity with "multiband corrections and lattice-regularization effects".

11 - In the 5th line of the caption of Fig. 1, replace "velocity" with "velocity $v$", so that the symbol $v$ has been defined before it appears again three lines below. Similarly, need to define $(\Delta v)_i$ below Eq. (2).

12 - Is the "static photoinduced spin magnetization" the same as the inverse Faraday effect? Is it correct that it does not require absorption/dissipation, while spin injection does?

13 - Miscellaneous typos: Second line of Sec. 2: "injectin". Below Eq. (2), after $\omega_{ln}=\varepsilon_l-\varepsilon_n$ there should be a comma, not a period.

14 - Replace the arXiv preprint in Ref. [26] by the published article.

15 - Fix capitalization in the article titles in the bibliography. For example, "taas" vs "TaAs" in Ref. [9], and many other similar issues. In bibtex, this can be achieved by placing the title entry inside double curly brackets.

16 - The text could benefit from one more round of polishing revisions. Some examples: "the three plots shows" (p. 3); "the changing rate" instead of "the rate of change" (p. 3), "can be heuristically described [as] in Fig. 1(a)" (p. 3); the title of Sec. 3 would read better as "Spin injection and the helicity of a Weyl node"; "but the expression of $\beta^\text{inj}_{ii}$" ("of" $\rightarrow$ "for") (p. 5); "located at the $k_z$ axis" $\rightarrow$ "located on the $k_z$ axis" (p. 5), etc.

---

## Round 2 · List of Changes

We would like to thank the referee for many helpful questions and comments. Below we give a point-by-point reply to referee’s comments.
“In Fig. 1(b), shouldn't the green (spin) arrows in hc all be pointing outwards, and those in hv all be pointing inwards ("hedgehog" pattern)?”
[REPLY] The hedgehog pattern is of course one example of the helicity. But generally speaking, a nontrivial helicity does not need to be tied to the hedgehog pattern. Here the pattern shown in Fig.1(b) actually corresponds to the model above Eq. 8, i.e., from a lattice model of the Weyl Hamiltonian. In our revised manuscript, we make this point clear.
“The caption of Fig. 1 refers to the "chirality" of light. Shouldn't it be "helicity", since it is referenced to the propagation direction?”
[REPLY] Both chirality and helicity are used for circular light in literatures. To avoid confusion, we use “circular polarization of light” in the caption instead in the revised manuscript.
“Figures 2 and 3 are almost identical, the only difference being that Fig. 3 contains one additional curve in panels (b,c,d). Would it make sense to simply replace Fig. 2 with Fig. 3, and change the text accordingly? (Possibly expand the last paragraph of Sec. 4 to include some of the material in the last paragraph of Appendix C)”
[REPLY] We thank the referee for this suggestion. In our revised manuscript, we replace Fig. 2 with Fig. 3 and expand the discussions.
“In both Figs. 2 and 3, the frequency range in panel (c) goes exactly from 0 to 3, whereas in panels (b,c) it goes slightly beyond 3, so that panels (b,d) are slightly "misaligned". Once they are aligned the tick labels above panel (b) could be removed, making the figure less busy.”
[REPLY] We thank the referee for this suggestion. We have modified the figure accordingly.
“Equation (10) is the same as (4), with an extra equality in the middle. Maybe replace Eq. (4) with (10), reducing by one the total number of equations?”
[REPLY] We have made this replacement.
“In Appendix C, τ1 and τ2 are dummy time-integration variables, while τ0 is the relaxation time. Would it be more clear to rename τ1 and τ2 as t1 and t2?”
[REPLY] We have made this replacement of labels.
“The relaxation time τ0 appears out of nowhere in Eq. (41). The way it is usually introduced is in the adiabatic switching on of the coupling to light. But do I understand correctly that then it should always appear in the combination ω+i/τ0? That does not seem to be the case in the denominator of Eq. (41). Also, in the definition of Gln below Eq. (41) is it really just ω→−ω, or should the sign change affect i/τ0 as well?”
[REPLY] As stated by the referee, the relaxation time appears due to the adiabatic switching. Mathematically, it appears as the imaginary part of the frequency as pointed out the referee. It always appear in the combination of frequencies. The reason is that, for positive and negative frequency, their imaginary part has the same sign due to the causality.
In the revised manuscript, we have added discussions about the imaginary part of the frequency.
“Slightly inconsistent notation throughout the text concerning the traced quantity.”
[REPLY] We have made the notation consistent in the revised manuscript.
“The title refers to "spin injection", but the abstract talks about "injection spin", and both forms are used in the main text. Use consistently the first form?”
[REPLY] We use the term “spin injection” in the revised manuscript.
“In the abstract, "multiband and lattice effect" should be replaced for clarity with "multiband corrections and lattice-regularization effects".”
[REPLY] We thank the referee for this suggestion. We have made the replacement in the revised manuscript.
“In the 5th line of the caption of Fig. 1, replace "velocity" with "velocity v", so that the symbol v has been defined before it appears again three lines below. Similarly, need to define (Δv)i below Eq. (2).”
[REPLY] We have made the replacement in the revised manuscript.
“Is the "static photoinduced spin magnetization" the same as the inverse Faraday effect? Is it correct that it does not require absorption/dissipation, while spin injection does?”
[REPLY] The spin injection is one mechanism of the inverse Faraday effect and as stated by the referee, it involves the absorption. It can generalize a static photoinduced spin magnetization after reaching the steady state. There is a similar phenomenon: the circular photogalvanic effect. It generates an injecting current which is static at the steady state.
“Miscellaneous typos: Second line of Sec. 2: "injectin". Below Eq. (2), after ωln=εl−εn there should be a comma, not a period.
Replace the arXiv preprint in Ref. [26] by the published article.
Fix capitalization in the article titles in the bibliography. For example, "taas" vs "TaAs" in Ref. [9], and many other similar issues. In bibtex, this can be achieved by placing the title entry inside double curly brackets.
The text could benefit from one more round of polishing revisions. Some examples: "the three plots shows" (p. 3); "the changing rate" instead of "the rate of change" (p. 3), "can be heuristically described [as] in Fig. 1(a)" (p. 3); the title of Sec. 3 would read better as "Spin injection and the helicity of a Weyl node"; "but the expression of βinj ii" ("of" → "for") (p. 5); "located at the kz axis" → "located on the kz axis" (p. 5), etc.”
[REPLY] We thank the referee for many helpful suggestions. We have made the revision accordingly.
“In Fig. 1(b), shouldn't the green (spin) arrows in hc all be pointing outwards, and those in hv all be pointing inwards ("hedgehog" pattern)?”
[REPLY] The hedgehog pattern is of course one example of the helicity. But generally speaking, a nontrivial helicity does not need to be tied to the hedgehog pattern. Here the pattern shown in Fig.1(b) actually corresponds to the model above Eq. 8, i.e., from a lattice model of the Weyl Hamiltonian. In our revised manuscript, we make this point clear.
“The caption of Fig. 1 refers to the "chirality" of light. Shouldn't it be "helicity", since it is referenced to the propagation direction?”
[REPLY] Both chirality and helicity are used for circular light in literatures. To avoid confusion, we use “circular polarization of light” in the caption instead in the revised manuscript.
“Figures 2 and 3 are almost identical, the only difference being that Fig. 3 contains one additional curve in panels (b,c,d). Would it make sense to simply replace Fig. 2 with Fig. 3, and change the text accordingly? (Possibly expand the last paragraph of Sec. 4 to include some of the material in the last paragraph of Appendix C)”
[REPLY] We thank the referee for this suggestion. In our revised manuscript, we replace Fig. 2 with Fig. 3 and expand the discussions.
“In both Figs. 2 and 3, the frequency range in panel (c) goes exactly from 0 to 3, whereas in panels (b,c) it goes slightly beyond 3, so that panels (b,d) are slightly "misaligned". Once they are aligned the tick labels above panel (b) could be removed, making the figure less busy.”
[REPLY] We thank the referee for this suggestion. We have modified the figure accordingly.
“Equation (10) is the same as (4), with an extra equality in the middle. Maybe replace Eq. (4) with (10), reducing by one the total number of equations?”
[REPLY] We have made this replacement.
“In Appendix C, τ1 and τ2 are dummy time-integration variables, while τ0 is the relaxation time. Would it be more clear to rename τ1 and τ2 as t1 and t2?”
[REPLY] We have made this replacement of labels.
“The relaxation time τ0 appears out of nowhere in Eq. (41). The way it is usually introduced is in the adiabatic switching on of the coupling to light. But do I understand correctly that then it should always appear in the combination ω+i/τ0? That does not seem to be the case in the denominator of Eq. (41). Also, in the definition of Gln below Eq. (41) is it really just ω→−ω, or should the sign change affect i/τ0 as well?”
[REPLY] As stated by the referee, the relaxation time appears due to the adiabatic switching. Mathematically, it appears as the imaginary part of the frequency as pointed out the referee. It always appear in the combination of frequencies. The reason is that, for positive and negative frequency, their imaginary part has the same sign due to the causality.
In the revised manuscript, we have added discussions about the imaginary part of the frequency.
“Slightly inconsistent notation throughout the text concerning the traced quantity.”
[REPLY] We have made the notation consistent in the revised manuscript.
“The title refers to "spin injection", but the abstract talks about "injection spin", and both forms are used in the main text. Use consistently the first form?”
[REPLY] We use the term “spin injection” in the revised manuscript.
“In the abstract, "multiband and lattice effect" should be replaced for clarity with "multiband corrections and lattice-regularization effects".”
[REPLY] We thank the referee for this suggestion. We have made the replacement in the revised manuscript.
“In the 5th line of the caption of Fig. 1, replace "velocity" with "velocity v", so that the symbol v has been defined before it appears again three lines below. Similarly, need to define (Δv)i below Eq. (2).”
[REPLY] We have made the replacement in the revised manuscript.
“Is the "static photoinduced spin magnetization" the same as the inverse Faraday effect? Is it correct that it does not require absorption/dissipation, while spin injection does?”
[REPLY] The spin injection is one mechanism of the inverse Faraday effect and as stated by the referee, it involves the absorption. It can generalize a static photoinduced spin magnetization after reaching the steady state. There is a similar phenomenon: the circular photogalvanic effect. It generates an injecting current which is static at the steady state.
“Miscellaneous typos: Second line of Sec. 2: "injectin". Below Eq. (2), after ωln=εl−εn there should be a comma, not a period.
Replace the arXiv preprint in Ref. [26] by the published article.
Fix capitalization in the article titles in the bibliography. For example, "taas" vs "TaAs" in Ref. [9], and many other similar issues. In bibtex, this can be achieved by placing the title entry inside double curly brackets.
The text could benefit from one more round of polishing revisions. Some examples: "the three plots shows" (p. 3); "the changing rate" instead of "the rate of change" (p. 3), "can be heuristically described [as] in Fig. 1(a)" (p. 3); the title of Sec. 3 would read better as "Spin injection and the helicity of a Weyl node"; "but the expression of βinj ii" ("of" → "for") (p. 5); "located at the kz axis" → "located on the kz axis" (p. 5), etc.”
[REPLY] We thank the referee for many helpful suggestions. We have made the revision accordingly.

---

## Editorial Decision

published